# Effects of Ultrasound on Zinc Oxide/Vermiculite/Chlorhexidine Nanocomposite Preparation and Their Antibacterial Activity

**DOI:** 10.3390/nano9091309

**Published:** 2019-09-13

**Authors:** Karla Čech Barabaszová, Sylva Holešová, Kateřina Šulcová, Marianna Hundáková, Barbora Thomasová

**Affiliations:** Nanotechnology Centre, VSB-Technical University of Ostrava, 17. listopadu 15/2172, 708 00 Ostrava-Poruba, Czech Republic; sylva.holesova@vsb.cz (S.H.); katerina.sulcova.st@vsb.cz (K.Š.); marianna.hundakova@vsb.cz (M.H.); barbora.thomasova@vsb.cz (B.T.)

**Keywords:** ultrasonic intercalation, mechanical stirring, zinc oxide nanoparticles, vermiculite, chlorhexidine, antibacterial activity

## Abstract

Microbial infection and biofilm formation are both problems associated with medical implants and devices. In recent years, hybrid organic-inorganic nanocomposites based on clay minerals have attracted significant attention due to their application potential in the field of antimicrobial materials. Organic drug/metal oxide hybrids exhibit improved antimicrobial activity, and intercalating the above materials into the interlayer of clay endows a long-term and controlled-release behavior. Since antimicrobial activity is strongly related to the structure of the material, ultrasonic treatment appears to be a suitable method for the synthesis of these materials as it can well control particle size distribution and morphology. This study aims to prepare novel, structurally stable, and highly antimicrobial nanocomposites based on zinc oxide/vermiculite/chlorhexidine. The influence of ultrasonic treatment at different time intervals and under different intercalation conditions (ultrasonic action in a breaker or in a Roset’s vessel) on the structure, morphology, and particle size of prepared hybrid nanocomposite materials was evaluated by the following methods: scanning electron microscopy, X-ray diffraction, energy dispersive X-ray fluorescence spectroscopy, carbon phase analysis, Fourier transforms infrared spectroscopy, specific surface area measurement, particle size analysis, and Zeta potential analysis. Particle size analyses confirmed that the ultrasonic method contributes to the reduction of particle size, and to their homogenization/arrangement. Further, X-ray diffraction analysis confirmed that ultrasound intercalation in a beaker helps to more efficiently intercalate chlorhexidine dihydrochloride (CH) into the vermiculite interlayer space, while a Roset’s vessel contributed to the attachment of the CH molecules to the vermiculite surface. The antibacterial activity of hybrid nanocomposite materials was investigated on Gram negative (*Escherichia coli*, *Pseudomonas aeruginosa*) and Gram positive (*Staphylococcus aureus*, *Enterococcus faecalis*) bacterial strains by finding the minimum inhibitory concentration. All hybrid nanocomposite materials prepared by ultrasound methods showed high antimicrobial activity after 30 min, with a long-lasting effect and without being affected by the concentration of the antibacterial components zinc oxide (ZnO) and CH. The benefits of the samples prepared by ultrasonic methods are the rapid onset of an antimicrobial effect and its long-term duration.

## 1. Introduction

Currently, many different methods and processes (hydrothermal, chemical coprecipitation [1], sol-gel synthesis, and sonochemical and mechanochemical synthesis, etc.) are used for nanocomposite material preparation. Hybrid antibacterial nanocomposite materials are specific examples of nanocomposite materials, which are increasingly in demand, especially in the fields of medicine, biomedical, and food applications [2,3].

Hybrid nanomaterials, formed by two or more components with at least one component being at the nanometric dimension, combine the intrinsic characteristics of their individual constituents to give additional properties due to the synergistic effects between the components. The main reason why these individual components are connected together is to combine their most favorable properties with one another and at the same time to eliminate the disadvantages of each component [4]. The preparation of hybrid nanocomposite materials is itself very difficult and demanding. Complications are mainly caused by a large difference in the working temperatures of the individual organic and inorganic components. Organic nanomaterials are typically treated at ambient temperatures up to 200 °C, in contrast to inorganic nanomaterials that require higher temperatures. Therefore, the appropriate choice of method and preparation conditions is a key element in the final nature of the material [5,6].

The ultrasonic method is widely used as an efficient and practical way for the preparation of a huge range of materials with controlled properties. Ultrasonic treatment is considered to be a method where either the crystal structure is not or is only minimally disrupted [7]. The main advantage of ultrasonic treatment lies in the versatility by which nanostructured metals, oxides, chalcogenides, carbides, polymers, and nanocomposite materials can be prepared. Ultrasonic treatment is also important for the preparation of micron and submicron particles of layered clay minerals. Under the influence of ultrasound, the clay mineral is exposed to thermal shock and steam generation in the interlayer, and subsequent exfoliation of individual layers occurs. 2:1 phyllosilicates, which in an exfoliated state exhibit low bulk density, thermal conductivity, and a relatively high melting point, are most commonly used for this method. The size of the synthesized particles is closely related to the length of the processing time [8,9]. A change of working conditions can also have a significant effect on both the preparation process and the resulting character of the material. Efficient utilization of the Roset type reactor has been studied in the preparation of Na-vermiculite micron particles [10]. The most noticeable effect on particle size is the change in temperature, where, with increasing temperature (90 °C), the particle size and span of time rapidly decrease. An effective particle reduction process is due to exfoliation and delamination of particles as a result of cavitation and evaporating water bubbles at higher temperatures. Longer exposure to ultrasound under these conditions leads to particle size growth due to aggregation and subsequent deagglomeration. Ultrasonic treatment can also be effectively used for intercalation, e.g., polyethylene glycol into the montmorillonite structure. Intercalation also reduces the montmorillonite particles without destroying the crystal structure [11].

The development of suitable antibacterial materials that can be used for medical purposes is a current medical research topic. Substances that act as carriers of antibacterial or antifungal agents for topical treatment that avoid undesirable treatment of the entire body are of particular interest [12]. For practical applications, sustained antibacterial activity with regular drug release is desirable. Antibacterial agents can be divided into two categories, based on their chemical composition: organic and inorganic. Organic antibacterial materials are often less stable, especially at high temperatures and pressures, and they tend to evaporate or decompose. On the other hand, they possess organophilicity, so it is easy for them to adhere to and exterminate microbes. Inorganic antibacterial agents that have prolonged exposure times, chemical stability, and heat resistance are receiving increasing attention. In particular, metal ions and oxides of silver (Ag^+^), zinc (Zn^2+^), or copper (Cu^2+^), which effectively inhibit the birth and growth of harmful microbes, are currently used [13,14]. Sustained antibacterial activity with a regular release of nanomaterial drugs is also not negligible. The antibacterial activity of zinc oxide nanoparticles depends primarily on particle size, particle specific surface area, and concentration [15,16,17,18,19]. The most significant antibacterial activity of zinc oxide nanoparticles is reported against *Escherichia coli* and *Staphylococcus aureus* bacterial strains. Nowadays, clay minerals such as smectites (montmorillonite, saponite), talc, or kaolinite are frequent carriers of antibacterial agents (drugs). Layered silicates have become a part of biological systems due to their inertia and non-toxic character [20].

Vermiculite (2:1 layered silicates) is one of the naturally occurring clay silicate minerals with layers built up of one octahedral sheet sandwiched between two tetrahedral sheets. The central cations of octahedra (ideally Al^3+^) and tetrahedra (ideally Si^4+^) can be substituted by cations. These substitutions result in a negative layer charge on the silicate layers. Space between layers (interlayer space) is occupied by exchangeable hydrated cations such as Mg^2+^, Ca^2+^, Na^+^, K^+^, etc., compensating the negative layer charge. The magnitude of layer charge on the vermiculite layers has a key effect on the amount of intercalated organic molecules and on their arrangement in the interlayer space.

Vermiculite does not exhibit antibacterial activity [21] but does adsorb and kill bacteria as soon as their structure is intercalated by an antibacterial agent. Very good results of the antibacterial activity of chlorhexidine in a vermiculite structure have been demonstrated with a nanocomposite material that was prepared by an ion exchange reaction. The efficacy against *S. aureus* was confirmed by samples with the lowest chlorhexidine concentration, and the activity was proven even after prolonged exposure. The results also showed that even the highest chlorhexidine dihydrochloride (CH) concentration used did not lead to the complete intercalation and destruction of the vermiculite crystal structure [12].

In this work, we have investigated the preparation conditions of the hybrid nanocomposite based on zinc oxide/vermiculite/chlorhexidine derived by ultrasound treatment. The aim was to prepare a structurally stable and highly antibacterial nanomaterial with a defined concentration composition of organic and inorganic components.

## 2. Materials and Methods

### 2.1. Materials and Nanocomposites Preparation

Natural Mg-vermiculite from Brazil (supplied by Grena Co., Veselí nad Lužnicí, Czech Republic) was used as a starting material for the nanocomposite sample preparation. Natural vermiculite was milled in a planetary ball mill (Retsch PM4) and then sieved through a 0.040 mm sieve (sample named V).

The zinc oxide/vermiculite (ZnO/V) nanocomposite was prepared by the sonochemical method in 1 M NaCl solution followed by a heat treatment. The required amount of NaCl was dissolved in 100 mL of distilled water and heated on an electric heater to 80 °C. Five grams of vermiculite, 2.5 g of anhydrous ZnCl_2_, and 2.5 g of anhydrous Na_2_CO_3_ (all from Sigma Aldrich) were gradually added to the solution. The titanium sonotrode (UP100H from Hielscher, Teltow, Germany) was placed in the suspension and sonicated for 15 min throughout the cycle at 50% amplitude. Subsequently, the sample was washed with distilled water and centrifuged until chlorides disappeared. The solid nanocomposite material was dried and homogenized. The homogenized sample was calcined at 350 °C for 1.5 h. The sonochemically prepared inorganic nanocomposite sample was designated ZnO/V.

The ZnO/V/CH nanocomposites were prepared by the intercalation of chlorhexidine dihydrochloride (CH) using two different techniques—mechanical stirring and ultrasonic action.

During mechanical stirring, 2 g of the ZnO/V nanocomposite was mixed in 50 mL of demineralized water for 3 min and was then added to the 50 mL ethanol solution, in which 2 g of CH had been previously dissolved. The dispersions were homogenously mixed for 5 h at 75 °C on a magnetic stirrer. The final dispersion was freed from water by centrifugation and drying at 75 °C for 24 h. The hybrid nanocomposite material was named ZnO/V_M_CH.

Identical chemical precursors were used for ultrasonic intercalation. The ultrasonic titanium sonotrode was placed in the dispersion in a beaker, and the mixture was subjected to ultrasound treatment at 30 and 90 min intervals. The dispersion was subsequently centrifuged and dried at 75 °C for 24 h (hybrid nanocomposite samples were named ZnO/V_30UCH and ZnO/V_90UCH). Under the same conditions, dispersions were subjected to ultrasound treatment in a Roset’s vessel (hybrid nanocomposite samples were named ZnO/V_30UCH_R and ZnO/V_90UCH_R).

### 2.2. Analytical Methods and Equipment

The chemical composition of vermiculite (V) and the hybrid nanocomposite samples was obtained from elemental analysis by X-ray fluorescence spectroscopy (SPECTRO XEPOS new energy dispersive X-ray fluorescence spectrometer).

The morphology of the nanocomposite samples was investigated using scanning transmission electron microscope (STEM) JEOL JSM-7610F Plus, Tokyo, Japan. The samples were coated with a gold/palladium film in order to avoid problems with electrical charging. SEM images were obtained using a scattered electron detector (SE, LEI).

The X-ray powder diffraction (XRD) analysis was performed using the diffractometer RIGAKU Ultima IV (scintilation detector, CuK*α* radiation, NiK*β* filter, Bragg–Brentano arrangement, Tokyo, Japan. Samples in a standard holder were measured in ambient atmosphere (40 kV, 40 mA, 2.32°/min). Phase analysis was evaluated by database PDF-2 Release 2011.

The organic carbon content in the hybrid nanocomposite samples was determined using phase carbon analyzer RC612 (LECO, MI, USA). The defined weight of the hybrid nanocomposite samples was found by burning them in an oxygen atmosphere with a temperature range of 100 °C to 1000 °C. The carbon was detected in the IR cells in the form of CO_2_.

The IR spectra of the original V, ZnO/V, and hybrid nanocomposite samples were obtained using the potassium bromide pellets technique. Exactly 1.0 mg of each sample was ground with 200 mg of dried potassium bromide. This mixture was used to prepare the potassium bromide pellets. The pellets were pressed under 8 tons for 30 s under vacuum. The IR spectra were collected using FTIR (spectrometer Nexus 470 (ThermoScientific, MA, USA) with a DTGS detector. The measurement parameters were as follows: spectral region, 4000–400 cm^−1^; spectral resolution, 4 cm^−1^, 64 scans; Happ–Genzel anodization. Treatment of spectra: polynomial (second order) baseline, subtraction spectrum of pure potassium bromide.

The particle size of all samples was determined by the HORIBA Laser diffraction particle size analyser (LA-950 instrument, Kyoto, Japan) with a short-wavelength blue and red light source in conjunction with forward and backscatter detection. The particle size analyses were conducted with refractive indices of 1.54 (for vermiculite), 1.50 (for zinc oxide), 1.54 (for chlorhexidine), and 1.33 (for water).

Zeta potential (*ξ*-potential) was measured by a nanoparticle analyzer (HORIBA Nanopartica SZ-100, Kyoto, Japan) equipped with a microprocessor unit to directly calculate the *ξ*-potential. The system measures the sample conductivity, applies an electric field, and then measures the motion of the particles using electrophoretic light scattering. Each sample (0.5 g) was mechanically mixed with 50 mL of distilled water. One milliliter of the suspension was introduced into the disposable zeta potential cell. The *ξ*-potential was measured at natural pH. Each data point was an average of approximately 6 measurements. All measurements were made at ambient temperature (24.9 °C), conductivity (0.194 mS.cm^−1^), suspension viscosity (0.93 mPa.s), and electrode voltage constant (3.4 V). The *ξ*-potential was calculated using the Smoluchowski equation.

Specific surface area (SSA) was measured at liquid nitrogen atmosphere by means of a thermo scientific surfer. Prior to measurements, the samples were degassed under vacuum (10^−6^ bar) at 120 °C for 24 h. The SSA was calculated using the BET (Brunauer–Emmett–Teller) equation by assuming the area of the nitrogen molecule was 0.1620 m^2^. The total pore volume (PV) was obtained from the maximum amount of nitrogen gas adsorbed at partial pressure (*p/p*^0^) = 1.

### 2.3. Antibacterial Tests

The antibacterial activity of the samples was tested against four different human pathogenic strains; Gram negative strains (*Escherichia coli*, CCM 3954 and *Pseudomonas aeruginosa*, CCM 3955) and Gram positive strains (*Staphylococcus aureus*, CCM 3953 and *Enterococcus faecalis*, CCM 4224). The results were determined using a standard microdilution method, which enables the determination of the minimum inhibitory concentration (MIC) that completely inhibits bacterial growth in accordance with their lowest concentration.

The dilution and cultivation were performed on a microtitration plate with 96 hollows. The first set of hollows on the plate contained 10% (*w*/*v*) sample water dispersions. These dispersions were further diluted by a threefold diluting method in glucose stock in such a manner that the second to seventh sets of hollows contained samples dispersed in concentrations of 5%, 1.67%, 0.56%, 0.19%, 0.062%, 0.021%, and 0.007%. The eighth set of hollows contained pure glucose stock as a control test. A volume of 1 μL of glucose suspensions of *E. coli* (1.5 × 10^9^ CFU ml^−1^), *P. aeruginosa* (1.7 × 10^9^ CFU ml^−1^), *S. aureus* (1.7 × 10^9^ CFU ml^−1^), and *E. faecalis* (1.6 × 10^9^ CFU ml^−1^), provided by the Czech collection of microorganisms (CCM), was put into the hollows. Bacterial suspensions, after the elapse of 30, 60, 90, 120, 180, 240 and 300 min, and then over 2 days at 12 h intervals, were transferred from each hollow to 100 μL of the fresh glucose stock and bacteria, and were incubated in a thermostat at 37 °C for 24 and 48 h. Antibacterial activity was evaluated by turbidity, which is a display of bacterial growth.

## 3. Results

### 3.1. X-ray Fluorescence Analysis

The changes in the chemical composition from the elemental analysis of all experimental samples were calculated to the stoichiometric metal oxide concentrations and are summarized in Table 1. Mechanical and ultrasonic processing of nanocomposite particles leads to chemical changes, mostly in SiO_2_, Al_2_O_3_, Fe_2_O_3_, and MgO contents [22].

The initial weight composition of the ZnO in the sonochemically prepared ZnO/V nanocomposite was 23.57 wt%. The mechanical intercalation of CH caused a decrease in the ZnO composition to 20.44 wt% (ZnO/V_M_CH), while ultrasonic intercalation led to a decreased composition from 23.57 wt% to 14.87 wt% (ZnO/V_30U_CH) and to 12.87 wt% (ZnO/V_30U_CH_R). The chemical composition of the other oxides was also adequately reduced.

The values of organic carbon content (*f*_oc_) predicted and confirmed the level of CH intercalation in the hybrid nanocomposites (Table 1). The results show that the lowest carbon content (13.86 wt%) have mechanically intercalated the nanocomposite sample (ZnO/V_M_CH). Conversely, the highest *f*_oc_ value (27.62 wt%) was measured for the ZnO/V_30U_CH_R sample, prepared in a Roset’s vessel after 30 min of ultrasound treatment. Higher concentrations of chlorides in hybrid nanocomposites also confirmed a higher rate of intercalation by ultrasonic processes.

From XRFS analysis, it can be assumed that during the mechanical intercalation of CH weak mechanical/friction forces are exerted, which do not significantly affect the ZnO amount in the structure of the nanocomposite material (ZnO/V_M_CH). Conversely, the ultrasonic intercalation of CH due to the strong cavitation effect causes both surface disruption of the vermiculite particles and disruption of the bonds between the ZnO nanoparticles and V. It can be assumed that this process contributes to the emergence of new areas where higher amounts of CH are anchored.

### 3.2. Nanocomposite Particles Morphology

The morphological changes of nanocomposite samples are evident from the scanning transmission electron images (Figure 1). It is known that the natural V samples are formed by the particles with irregular shapes and sizes and smooth lamellar morphologies [22]. The ZnO nanoparticles in sonochemically prepared ZnO/V nanocomposites (Figure 1a) are widely dispersed on the surfaces and mainly decorate the V particle’s edges [20]. Smaller particle fractions, which are agglomerated together and are in contact with bigger fractions by the particle edges, are present in the volume. The ZnO/V nanocomposite particles intensively agglomerated.

The mechanical intercalation of CH (Figure 1b, sample ZnO/V_M_CH) had no significant effect on the particle morphology changes; the particles also formed agglomerates. It can be assumed that the CH produces the thin film that coats the original ZnO/V particles.

The ultrasonic intercalation of CH in a beaker produced the rounded and frayed edges of nanocomposite particles (Figure 1c,d). The bigger and smaller particle fractions, which are separated from each other, are evident. The particle surfaces are composed of individual platelets with sharp edges and are smooth without the evident presence of ZnO nanoparticles. The samples prepared in a Roset’s vessel (Figure 1e,f) produced particles with a representative lamellar structure and smooth surface. This can be attributed to the intense movement of nanocomposite particles in the liquid phase (water) and the washout/removal of ZnO nanoparticles from vermiculite surfaces. The CH caused clarification of vermiculite edges as a result of CH intercalation.

### 3.3. X-ray Diffraction Analysis

The XRD pattern of sample ZnO/V shows reflections with interlayer distance values of *d* = 1.224 nm, 1.135 nm, and 1.001 nm (Figure 2a and Figure 3a). The interlayer distance *d* = 1.224 nm signifies that ZnO nanoparticles can be incorporated into the vermiculite interlayer [23,24]. The *d* = 1.001 nm belongs to the dehydrated phase of the vermiculite structure and *d* = 1.135 nm indicates a different hydration state with a different amount of water molecules. The values *d* = 0.477 nm, 0.328 nm, 0.154 nm, and 0.444 nm belong to the vermiculite reflections. The reflections with *d* = 0.842 nm and 0.312 nm are ascribed to the admixture phase in vermiculite [25].

Reflections for the hexagonal wurzite structure of ZnO (PDF card no. 01-079-2205) were confirmed on the XRD-pattern of ZnO/V at 2*θ* = 31.80°, 34.17°, 36.22°, 47.60°, 56.56°, and 62.81°, and 68.06° corresponds to *d*-values 0.282 nm, 0.262 nm, 0.248 nm, 0.191 nm, 0.163 nm, 0.148 nm, and 0.138 nm.

After treatment of ZnO/V with CH by mechanical stirring (Figure 2a and Figure 3a), new reflections appeared for sample ZnO/V_M_CH. The interlayer distance values *d* = 2.96 nm, 2.158 nm, 1.649 nm, and 1.069 nm correspond to these reflections, which confirms the expansion of interlayer space of vermiculite due to intercalation of CH [12,26]. Moreover, the XRD pattern confirms the reflections of non-intercalated CH on the surface of the V substrate. ZnO reflections were also confirmed on the XRD pattern of ZnO/V_M_CH.

The XRD patterns of nanocomposites with CH intercalated by ultrasonic method (Figure 2b and Figure 3b) show reflections with interlayer distance values of *d* = 2.950 nm, 2.144 nm, 1.670 nm, 1.200, and 1.054 nm. There are only very small shifts in the comparison with ZnO/V_M_CH, which may be related to the arrangement of molecules in the interlayer as a result of the lower content of interlayer cations after ultrasonic intercalation, which was confirmed by XRFS analysis.

Based on the intensity of CH reflections on the vermiculite surface, the samples with CH intercalated by ultrasonic method contained the higher amounts of CH on the vermiculite surface in comparison with ZnO/V_M_CH. These results also correspond with the *f*_oc_ analysis, which confirmed a higher amount of *f*_oc_ in these samples. The ultrasonic action can cause greater disruption of the vermiculite surface structure due to the strong cavitation action, contributing to the formation of new areas on which a higher amount of CH is attached.

Nanocomposites with CH intercalated by the ultrasonic method for 30 min or 90 min in two types of vessels, the beaker and the Roset’s vessel, did not show significant differences. Samples prepared for 90 min showed higher relative intensity compared with those prepared for 30 min. Samples prepared in the Roset’s vessel showed lower relative intensity compared with samples prepared in the beaker.

The crystallite size (*L*_c_) of ZnO in the samples was calculated based on (101) reflection (about 36.2° 2*θ*) according the Scherrer’s equation [27]. The highest *L*_c_ value showed in sample ZnO/V (8.41 nm). After mechanical intercalation of CH, the *L*_c_ value of ZnO decreased to 7.71 nm for ZnO/V_M_CH. After ultrasonic intercalation of CH, the *L*_c_ value of ZnO decreased to 7.11 nm for ZnO/V_30U_CH, ZnO/V_30U_CH_R, and ZnO/V_90U_CH_R and to 7.05 nm for ZnO/V_30U_CH.

### 3.4. FTIR Spectroscopy

The IR spectrum of ZnO/V (Figure 4) shows bands at 3698 and 3439 cm^−1^ in the OH stretching region attributed to structural OH groups of V, and adsorbed water there to OH bending vibration at 1631 cm^−1^ also characterizes adsorbed water. The intensive band at 1002 cm^−1^ is assigned to Si–O stretching vibration, together with Si–O bending vibration at 453 cm^−1^, which overlaps the Zn–O band that occurs in a similar position [24,28].

Since spectra of all prepared ZnO/V/CH nanocomposites show vibrations characterizing the presence of CH at almost the same positions, differing only by intensities, we selected two samples where CH was intercalated by the mechanical method (ZnO/V_M_CH, Figure 4) and one by the ultrasonic methods (ZnO/V_90U_CH, Figure 4). The presence of CH in these nanocomposites was confirmed by the following vibrations: 3316 (3311), 3200 (3196), and 3120 (3124) cm^−1^, which correspond to NH stretching vibrations of secondary amine and imine functional groups, and by two bands at 2937 (2938) and 2858 (2856) cm^−1^, which are assigned to asymmetric and symmetric C–H stretching bands of CH. C–N stretching vibration of the imine group appeared at 1637 (1636) cm^−1^. The bands occurring at region 1604–1416 cm^−1^ originate from N–H bending vibration of secondary amine and imine groups and further from the stretching C–C vibrations of aromatic rings. Finally, bands at 824 cm^−1^ belong to the C–H out-of-plane deformation rocking vibration of methylene groups [24,29].

Figure 5 shows comparisons of band intensities in the FTIR spectra of ZnO/V/CH nanocomposites prepared via different methods. In the region 3500–1100 cm^−1^, mainly including C–H and N–H stretching and bending vibrations of CH, we can observe increasing intensities of the characteristic CH bands from samples prepared by the mechanical method (ZnO/V_M_CH) compared to those prepared by the ultrasonic methods, which is in very good agreement with results from analysis of total organic carbon content. Moreover, in the region below 1100 cm^−1^, where there are predominantly characteristic bands of V, we can observe a decrease in the characteristic V bands from samples prepared via mechanical method compared to those prepared by ultrasonic methods.

### 3.5. Particle Size and Surface Characteristics

The particle size distributions (PSD) in the volume content are shown in Figure 6 for all experimental samples. The particle size (PS) parameters were measured by the laser diffraction method in liquid mode. The PS parameters (Table 2), obtained from the particle size distribution data, were: volume-weighted mean diameter (De Brouckere mean diameter (*d*_43_) and mode diameter (*d*_m_)), which corresponds to the maximum peak of the frequency distribution.

The particle size distributions of the samples show modal and bimodal characteristics. The narrow monomodal PSD with a mean diameter of *d*_43_ = 13.25 μm was measured for the natural V sample, which was used as a starting material (Figure 6a). The sonochemically prepared ZnO/V nanocomposite sample led to a decrease in the PS (*d*_43_ = 9.07 µm), but it gave rise to a bimodal characteristic of PSD. However, a very narrow PSD in the area of higher size fractions was retained. This confirms the presence of two size fractions with mode diameters of *d*_m_ = 0.25 µm and *d*_m_ = 10.10 µm. These results correspond to the structural changes on the V particles (as seen in the SEM images, Figure 1).

From a particle size analysis point of view, mechanical intercalation of CH had an effect on the particle behavior in the sample volume (ZnO/V_M_CH, Figure 6a). The PSD had a bimodal characteristic with two mode diameters of 0.39 µm and 15.17 µm (*d*_m_), whereby interaction in higher size fractions (particle agglomeration) is evident. While the distribution range for the ZnO/V sample was in the range of 2.60 µm to 34.26 µm (Figure 6a), after mechanical intercalation of CH, the interval is much wider; from 1.15 µm to 300.52 µm (ZnO/V_M_CH, Figure 6a). It corresponds to the SEM images, where agglomeration and/or coagulation behavior were confirmed.

The ultrasonic intercalation of CH resulted in the reduction and arrangement of PS to the mean diameter *d*_43_ = 5.83 µm for ZnO/V_30U_CH (Figure 6b) and *d*_43_ = 6.69 µm for ZnO/V_90U_CH_R, and did not affect the bimodal character of the PSD. Simultaneously, any effects of ultrasonic exposure time and the influence of the preparation conditions (in the beaker or the Roset’s vessel) on the PS and PSD changes were not noticed.

The values of specific surface area (SSA) and the total pore volume (PV) were measured by nitrogen gas-adsorption and are given in Table 2. The SSA of natural V is high, 93.18 m^2^·g^−1^, and corresponds to reported values measured by Valášková, M. et al. [21,22]. The SSA rapidly decreased with the sonochemical preparation of the ZnO/V sample, and with the next mechanical intercalation of CH, to a value around 22.5 m^2^**·**g^−1^. The hybrid nanocomposite samples with intercalated CH show a similar specific surface area in interval values from 14.46 m^2^**·**g^−1^ (ZnO/V_30U_CH_R) to 17.69 m^2^**·**g^−1^ (ZnO/V_30U_CH) and reached a maximum of 17.69 m^2^**·**g^−1^ (ZnO/V_30U_CH). This decrease directly correlated to the reduction in particle size and particle size distribution. This phenomena also corresponds to the PV results where decreased characteristics between mechanically and ultrasonically intercalation of CH is evident.

The intercalation of CH was realized on negatively charged surfaces with the *ξ*-potential −20.6 mV (ZnO/V sample) and led to hybrid nanocomposites with positively charged surfaces (Table 2). The differences in the *ξ*-potential of mechanically intercalated CH in the ZnO/V_M_CH sample (27.7 mV) and ultrasonic intercalation of CH (*ξ*-potential from 23.4 mV to 23.9 mV) are evident. They are in agreement with the elemental changes in the sample structures (Table 1), mainly with the decreasing of ZnO, SiO_2_, Al_2_O_3_, and MgO and the increasing of CH. These findings correspond with the knowledge [8] that the position of the central cations in octahedra (ideally Al^3+^) and tetrahedra (ideally Si^4+^) of vermiculites are substituted by ZnO and CH, and result in the positively charged surfaces of hybrid nanocomposites.

### 3.6. Antibacterial Activity

Results of antibacterial activity (AC) measured at time intervals from 30 min to 1 day (at various time periods) are summarized for Gram negative strains (*E. coli* and *P. aeruginos*) in Table 3 and for Gram positive strains (*S. aureus* and *E. faecalis*) in Table 4. The AC was evaluated by finding the minimum inhibitory concentration (MIC).

The AC of the ZnO nanoparticles [18] and ZnO/V nanocomposite samples [30], with consideration of various conditions of preparation, was studied. It was found that these nanomaterials reduced the bacterial viability of *S. aureus* and *E. coli* and showed strong antibacterial activity, which was fast and long-acting after 120 min for more than 5 days. However, in this study, when samples were prepared under ultrasound, the ZnO/V sample was in all cases bacteria inactive (Table 3 and Table 4).

All hybrid nanocomposite samples prepared by the ultrasound method showed great antibacterial activity at the beginning of antibacterial testing (after 30 min) against both Gram negative strains and also against *S. aureus*. Moreover, in the case of *E. coli*, the best results were reached with both samples prepared in a Roset’s vessel (0.062% *w*/*v* MIC after 30 min, Table 3). Slightly worse results after 30 min were observed against *E. faecalis* (1.67% *w*/*v* MIC after 30 min, Table 4).

After one day of testing, almost all samples against all four strains showed the best antimicrobial efficiency (0.007% *w*/*v* MIC, Table 3 and Table 4). Some samples were slightly worse against very resistant *P. aeruginosa* (0.021% *w*/*v* MIC, Table 3).

Generally, sample ZnO/V_M_CH, prepared by the mechanical method, possessed slightly worse AC. It is probably connected with results from the XRD analysis, which showed us that this sample had less non-intercalated CH than the others prepared by the ultrasound method. For this reason, bacteria have worse contact with antimicrobial agents.

## 4. Conclusions

Antibacterial hybrid nanocomposite materials based on zinc oxide/vermiculite/chlorhexidine (ZnO/V/CH) with different concentrations of the organic component (chlorhexidine dihydrochloride) were prepared under different conditions using the ultrasonic method. Morphological, phase, elemental, and surface changes were compared with mechanically prepared hybrid nanocomposite materials. Phase analysis and structure analysis were performed by X-ray diffraction (XRD) and infrared spectroscopy (FTIR). The presence of CH and ZnO was confirmed in all samples. XRD analysis confirmed that ultrasound intercalation in a beaker helps more efficiently with the intercalation of CH to the vermiculite (V) interlayer space, while a Roset’s vessel contributed to the attachment of the CH molecules to the vermiculite surface. FTIR and total carbon analysis (*f*_oc_) showed that longer intercalation led to higher concentrations of CH in the structure.

All prepared hybrid nanocomposite materials showed the presence of a high percentage of total organic carbon content, while the highest values were measured in samples intercalated by the ultrasonic method. Particle size analyses confirmed that the ultrasonic method contributed to particle size reduction and their homogenization/arrangement. Particle size reduction during the ultrasonic intercalation was higher than during the mechanical intercalation (ZnO/V_M_CH). The ultrasonic intercalation of CH led to the decreasing of the specific surface area (SSA) and to the preparation of the particles with sharp edges as a result of CH intercalation to the interlayer space of the vermiculite particles. Increasing the concentration of CH decreased the SSA values, and the particles were more compact. The intercalation in a Roset’s vessel produced particles with lamellar structures and smooth surfaces.

The antibacterial activity of all hybrid nanocomposite materials has been demonstrated against *S. aureus*, *E. faecalis*, *E. coli*, and *P. aeruginosa*. Efficiency with very low MIC concentrations was observed after 30 min of exposure. The best results in this short time interval were against *E. coli* and came from samples prepared by the ultrasonic method in a Roset’s vessel.

The benefit of all the samples prepared by ultrasonic method is the rapid onset of an antimicrobial effect and its long-term duration. For that reason, the ultrasonic method represents a perceptive and efficient method of antibacterial hybrid nanocomposite material preparation. Compared to conventional intercalation procedures, it is a time-saving method enabling regulation of organic matter in the structure.

## Figures and Tables

**Figure 1 nanomaterials-09-01309-f001:**
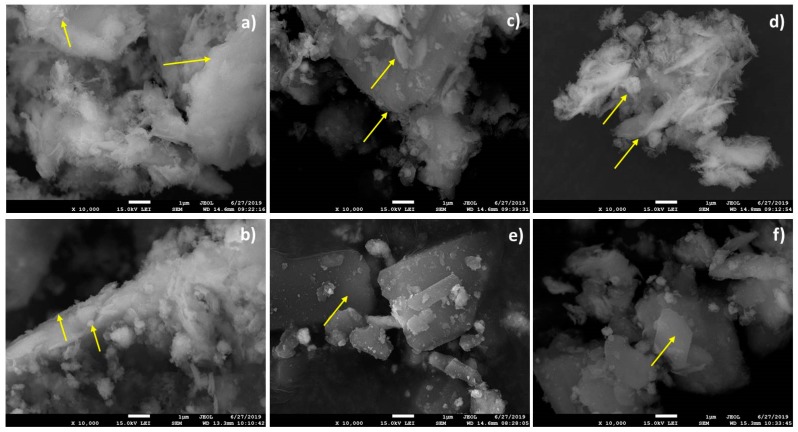
STEM images of the ZnO/V (**a**) and hybrid nanocomposite samples: (**b**) ZnO/V_M_CH, (**c**) ZnO/V_30U_CH, (**d**) ZnO/V_90U_CH, (**e**) ZnO/V_30U_CH_R, and (**f**) ZnO/V_90U_CH_R.

**Figure 2 nanomaterials-09-01309-f002:**
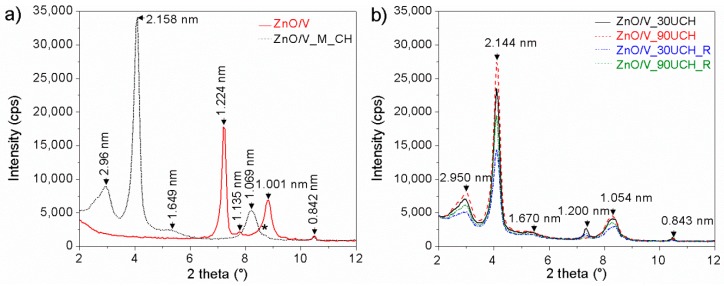
XRD patterns (from 2° to 12° 2θ) of samples (**a**) ZnO/V and ZnO/V_M_CH and (**b**) ZnO/V_30U_CH, ZnO/V_90U_CH, ZnO/V_30U_CH_R and ZnO/V_90U_CH.

**Figure 3 nanomaterials-09-01309-f003:**
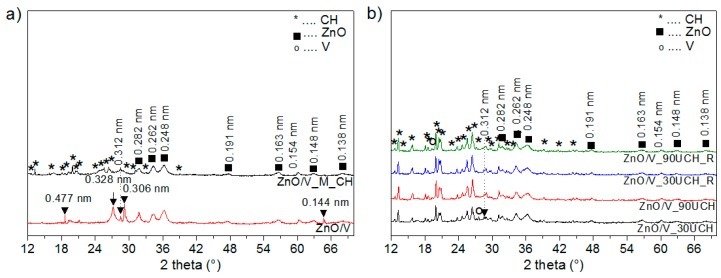
XRD patterns (from 12° to 70° 2θ) of hybrid nanocomposite samples (**a**) ZnO/V and ZnO/V_M_CH and (**b**) ZnO/V_30U_CH, ZnO/V_90U_CH, ZnO/V_30U_CH_R and ZnO/V_90U_CH.

**Figure 4 nanomaterials-09-01309-f004:**
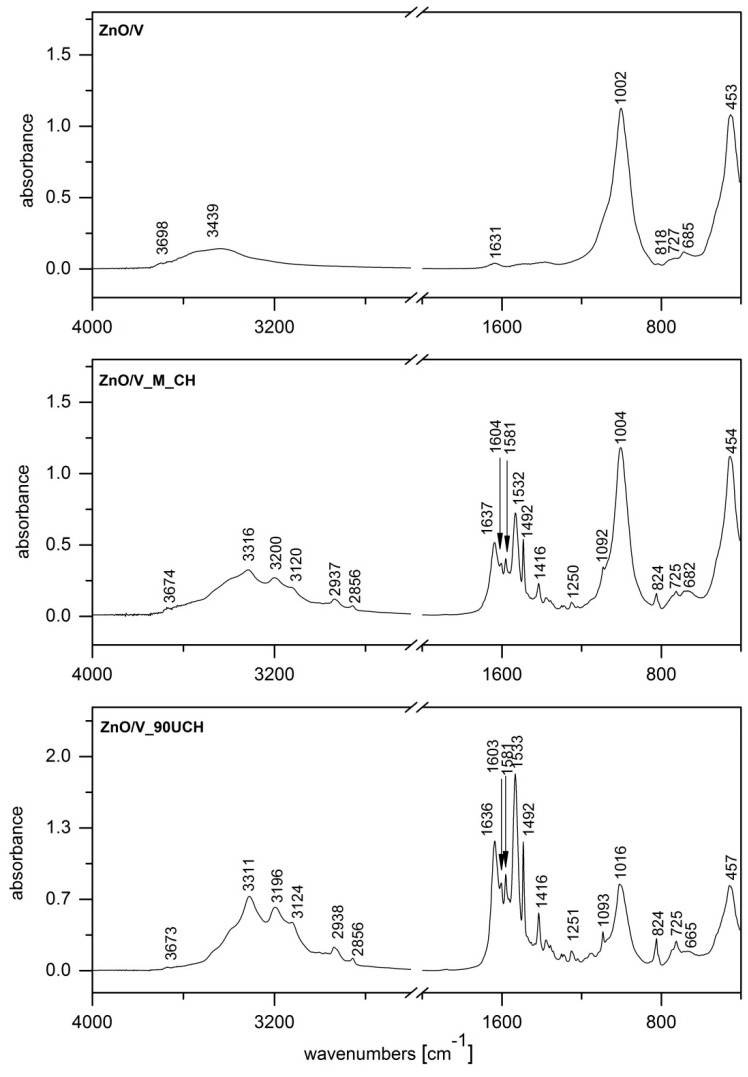
FTIR spectra of ZnO/V, ZnO/V_M_CH, and ZnO/V_90UCH.

**Figure 5 nanomaterials-09-01309-f005:**
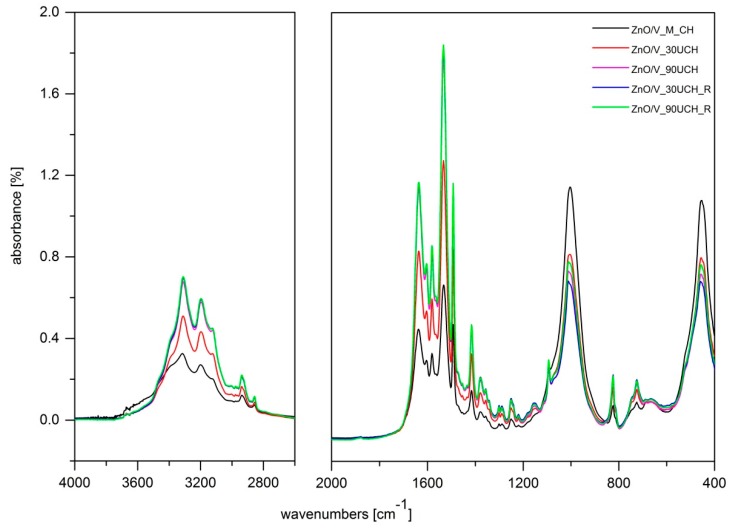
Comparison of band intensities—FTIR spectra of ZnO/V/CH nanocomposites prepared by different methods.

**Figure 6 nanomaterials-09-01309-f006:**
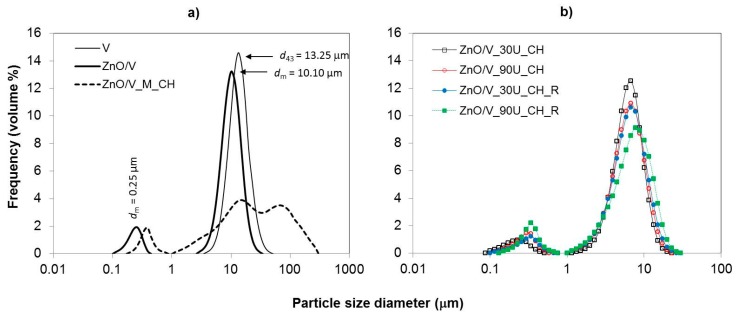
Log-normal particle size distribution of the (**a**) V, ZnO/V, and ZnO/V_M_CH and (**b**) ZnO/V_30U_CH, ZnO/V_90U_CH, ZnO/V_30U_CH_R, and ZnO/V_90U_CH_R nanocomposite samples.

**Table 1 nanomaterials-09-01309-t001:** The chemical composition (wt%) and organic carbon content (*f*_oc_) of the V, ZnO/V, ZnO/V_M_CH, ZnO/V_30U_CH, ZnO/V_90U_CH, ZnO/V_30U_CH_R, and ZnO/V_90U_CH_R nanocomposite samples.

Samples	V	ZnO/V	ZnO/V_M_CH	ZnO/V_30U_CH	ZnO/V_90U_CH	ZnO/V_30U_CH_R	ZnO/V_90U_CH_R
*f* _oc_	-	-	13.86	22.59	27.44	27.62	26.42
ZnO	-	23.57	20.44	14.87	13.82	12.87	13.74
SiO_2_	40.70	33.54	23.66	18.52	14.7	14.48	15.83
TiO_2_	1.10	0.84	0.78	0.53	0.51	0.45	0.50
Al_2_O_3_	10.10	8.50	7.21	6.66	5.74	5.97	6.18
Fe_2_O_3_	9.28	6.56	5.22	3.54	3.38	3.08	3.30
CaO	1.75	1.00	0.69	0.50	0.43	0.36	0.45
MgO	17.80	17.64	12.87	12.80	10.03	11.58	12.37
MnO	0.10	0.09	0.07	0.05	0.05	0.04	0.04
Na_2_O	0.60	<1	<1	<1	<1	<1	<1
K_2_O	2.76	1.85	1.49	1.02	0.95	0.86	0.91
Cl	-	0.10	7.32	12.75	16.51	16.50	13.85

**Table 2 nanomaterials-09-01309-t002:** Particle size and surface parameters.

Samples	*d*_m_ [μm]	*d*_43_ [μm]	SSA [m^2^·g^−1^]	PV [cm^3^·g^−1^]	*ξ*-Potential [mV]
V	13.25	13.48	93.18	0.107	−60.0
ZnO/V	0.25 10.10	9.07	22.99	0.030	−20.6
ZnO/V_M_CH	0.39 15.17	39.37	22.02	0.031	27.7
ZnO/V_30U_CH	0.23 6.72	5.83	17.69	0.026	23.4
ZnO/V_90U_CH	0.30 6.72	5.89	15.23	0.020	23.9
ZnO/V_30U_CH_R	0.34 6.72	6.19	14.46	0.026	23.4
ZnO/V_90U_CH_R	0.34 7.70	6.69	14.49	0.024	23.9

**Table 3 nanomaterials-09-01309-t003:** Antibacterial tests. Minimum inhibitory concentration (MIC) values (% *w*/*v*) of the experimental samples against *E. coli* and *P. aeruginosa* strains.

Samples	*E. coli* (MIC)				*P. aeruginosa* (MIC)			
	30 Min	120 Min	240 Min	1 Day	30 Min	120 Min	240 Min	1 Day
ZnO/V	-	-	-	-	-	-	-	-
ZnO/V_M_CH	0.56	0.19	0.19	0.007	0.56	0.19	0.19	0.021
ZnO/V_30U_CH	0.56	0.19	0.062	0.007	0.56	0.19	0.062	0.007
ZnO/V_90U_CH	0.19	0.062	0.062	0.007	0.19	0.19	0.19	0.007
ZnO/V_30U_CH_R	0.062	0.062	0.062	0.007	0.19	0.062	0.021	0.021
ZnO/V_90U_CH_R	0.062	0.062	0.062	0.007	0.19	0.062	0.062	0.021

**Table 4 nanomaterials-09-01309-t004:** Antibacterial tests. MIC values (% *w*/*v*) of the experimental samples against *S. aureus* and *E. faecalis* strains.

Samples	*S. aureus* (MIC)				*E. faecalis* (MIC)			
	30 Min	120 Min	240 Min	1 Day	30 Min	120 Min	240 Min	1 Day
ZnO/V	-	-	-	-	-	-	-	-
ZnO/V_M_CH	0.56	0.56	0.56	0.007	1.67	0.56	0.56	0.007
ZnO/V_30U_CH	0.56	0.56	0.062	0.007	1.67	0.19	0.19	0.007
ZnO/V_90U_CH	0.19	0.062	0.062	0.007	1.67	0.56	0.56	0.007
ZnO/V_30U_CH_R	0.19	0.19	0.062	0.007	0.56	0.19	0.19	0.007
ZnO/V_90U_CH_R	0.19	0.19	0.19	0.007	1.67	0.19	0.19	0.007

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
