# Peer review of "Effects of Ultrasound on Zinc Oxide/Vermiculite/Chlorhexidine Nanocomposite Preparation and Their Antibacterial Activity"

_nanomaterials, 2019, doi:10.3390/nano9091309_

Round 1

Reviewer 1 Report

This study by Barabaszová et al. investigates the effect of ultrasound on a hybrid nanocomposite system and looks into their antibacterial effect. While the study has presented some interesting findings, there are some major issues and shortcomings that need to be addressed to further improve the impact and significance of the work. Below are my specific comments: 

1- The Abstract seems to lack a proper structure, missing an important introductory statement describing why this study is significant. Why is this specific nanocomposite solution important? Why each specific compound was chosen in the composite system? Current Abstract is mostly description of Methods used. There is no intro and more importantly, no summary of results obtained. This section needs major work in order to attract and excite readers to continue reading this manuscript.

2- Figure 1 - the SEM results: The figure needs a more detailed caption/labelling to describe the experimental groups. "ZnO/V", "ZnO/V_30U_CH", and ... need to be defined in the caption. Also, I strongly suggest adding markers and arrows to highlight the key features in each panel. Otherwise, for a lay observer, it is difficult to distinguish any differences across groups/images.

3- Figures 2 and 3: would make more sense to combine these figures as a multi-panel figure, containing all XRD data. Same suggestion for Figures 4 and 5, for the FTIR data.

4- Figure 6: The inset in the figure needs to be described carefully. This is a general comment. Most figure captions in the manuscript lack detailed info about the components of each figure. I highly recommend authors to take the time to fully explain each figure, define study groups, and briefly touch on the main findings.

Author Response

Dear Walker Zheng,

we revised our manuscript on the basis of Reviewer's (1) comments appended below. We made revision with following supplements and/or changes and gave complementary reasons (text in red).

This study by Barabaszová et al. investigates the effect of ultrasound on a hybrid nanocomposite system and looks into their antibacterial effect. While the study has presented some interesting findings, there are some major issues and shortcomings that need to be addressed to further improve the impact and significance of the work. Below are my specific comments:

The Abstract seems to lack a proper structure, missing an important introductory statement describing why this study is significant. Why is this specific nanocomposite solution important? Why each specific compound was chosen in the composite system? Current Abstract is mostly description of Methods used. There is no intro and more importantly, no summary of results obtained. This section needs major work in order to attract and excite readers to continue reading this manuscript.

We totally rewrote part Abstract on the basis of yours comments, mainly in the point of view of significance and impact of the presented work and we also mentioned main results of our research.

Figure 1 - the SEM results: The figure needs a more detailed caption/labelling to describe the experimental groups. "ZnO/V", "ZnO/V_30U_CH", and ... need to be defined in the caption. Also, I strongly suggest adding markers and arrows to highlight the key features in each panel. Otherwise, for a lay observer, it is difficult to distinguish any differences across groups/images.

We added arrows to the each part in Figure 1 and we changed caption on the basis of yours comments which now correspond with description in the text.

Figures 2 and 3: would make more sense to combine these figures as a multi-panel figure, containing all XRD data. Same suggestion for Figures 4 and 5, for the FTIR data.

In the case of Figures 2 and 3 we had to make compromise between current state and multi-panel display. For the best illustration of situation during anchoring ZnO and CH on V it is necessary to display the enlarged detail of passage from 2 to 12 (2 Theta (°), Figure 2) and moreover to display all patterns together to compare their intensities, which would be meaningless when depicting each pattern separately. The remaining XRD data are plotted from 12 to 70 (2 Theta (°), Figure 3), when information in this passage tell us about presence of CH on surface of nanocomposites. If we display each pattern separately, moreover in full range, we could not see all useful information.

The Figure 4 has been redrawn to multi-panel figure.

The Figure 5, as shown, is illustrative to the readers for comparing the intensities of the spectra in selected regions, which correlates well with the results from total organic carbon analysis and like multi-panel figure would lose its meaning.

Figure 6: The inset in the figure needs to be described carefully. This is a general comment. Most figure captions in the manuscript lack detailed info about the components of each figure. I highly recommend authors to take the time to fully explain each figure, define study groups, and briefly touch on the main findings.

The Figure 6 was divided to two parts a) and b). Also captions were changed according the Figure 6.

Yours sincerely,

Karla Čech Barabaszová, et al

Reviewer 2 Report

The aim of the study was to obtain nanocomposite materials based on the zinc oxide, vermiculite and chlorhexidine. They were prepared by mechanically stirring and ultrasonic action in two steps. Antibacterial activity of hybrid nanocomposite materials was investigated on Gram negative and the Gram positive bacterial strains. All nanocomposite materials prepared by ultrasound methods showed high antibacterial activity.

The manuscript is an interesting study and scientifically of good quality, however, it has a character of simple technical report rather than scientific paper, because the results are simply arranged with the analysis flow. I would suggest changing the names of sections into names related to the described processes or problems.

Some specific comments:

Abstract: The abstract should emphasize the novelty of research more clearly. Also abbreviations should not be used in the abstract, the meaning of which is only explained in the article. Please limit them in the abstract and transfer them to the text if they are not there (such as XRFS). Line 26 and 27: Italics for Latin bacterial names is missing. Please correct. Line 136: “V”; an explanation of the meaning (abbreviation) should be given on the first use. It is unnecessary here. XRFS analysis: Authors should explain the obtained results and not just present them. Why mechanical intercalation of CH caused a greater reduction in composition compared to ultrasonic intercalation. Also for the X-ray diffraction analysis there should be more explanations of the effects observed. For example why occurred the expanding of interlayer space of vermiculite due to intercalation of CH for mechanical stirring. Line 314: Closing parenthesis is missing.

Author Response

Dear Walker Zheng,

we revised our manuscript on the basis of Reviewer's (2) comments appended below. We made revision with following supplements and/or changes and gave complementary reasons (text in red).

The aim of the study was to obtain nanocomposite materials based on the zinc oxide, vermiculite and chlorhexidine. They were prepared by mechanically stirring and ultrasonic action in two steps. Antibacterial activity of hybrid nanocomposite materials was investigated on Gram negative and the Gram positive bacterial strains. All nanocomposite materials prepared by ultrasound methods showed high antibacterial activity.

The manuscript is an interesting study and scientifically of good quality, however, it has a character of simple technical report rather than scientific paper, because the results are simply arranged with the analysis flow. I would suggest changing the names of sections into names related to the described processes or problems.

Some specific comments:

Abstract: The abstract should emphasize the novelty of research more clearly. Also abbreviations should not be used in the abstract, the meaning of which is only explained in the article. Please limit them in the abstract and transfer them to the text if they are not there (such as XRFS). Line 26 and 27: Italics for Latin bacterial names is missing. Please correct. Line 136: “V”; an explanation of the meaning (abbreviation) should be given on the first use. It is unnecessary here.

We rewrote part Abstract on the basis of yours comments, mainly in the point of view of significance and impact of the presented work and we also mentioned main results of our research. Further, Italics for bacterial strains names were used and we removed detailed description of sample V in mentioned line, we used only abbreviation “V” explained before in the text.

XRFS analysis: Authors should explain the obtained results and not just present them. Why mechanical intercalation of CH caused a greater reduction in composition compared to ultrasonic intercalation.

We added following comments about XRFS analysis to the text:

From XRFS analysis it can be assumed that during the mechanical intercalation of CH a weak mechanical / friction forces are exerted, which does not significantly affect the ZnO amount in the structure of the nanocomposite material (ZnO/V_M_CH). Conversely, the ultrasonic intercalation of CH due to the strong cavitation effect causes both surface disruption of the vermiculite particles and disruption of the bonds between the ZnO nanoparticles and V. It can be assumed that this process contributes to the emergence of new areas where higher amounts of CH are anchored.

Also for the X-ray diffraction analysis there should be more explanations of the effects observed. For example why occurred the expanding of interlayer space of vermiculite due to intercalation of CH for mechanical stirring.

Please, find explanation in the text which was completed.

Line 314: Closing parenthesis is missing.

It was corrected.

Yours sincerely,

Karla Čech Barabaszová, et al

Round 2

Reviewer 2 Report

Now is OK. Thank you